# Evaluation of the Nasolabial Angle in Orthodontic Diagnosis: A Systematic Review

**Vincenzo Quinzi [1]** , **Licia Coceani Paskay [2]**, **Nicola D'Andrea [1]**, **Arianna Albani [1]**, **Annalisa Monaco [1]** and **Sabina Saccomanno [1],\***

1   Department of Health, Life and Environmental Sciences, University of L'Aquila, 67100 L'Aquila, Italy; vincenzo.quinzi@univaq.it (V.Q.); nicola.dandrea@student.univaq.it (N.D.); arianna.albani@student.univaq.it (A.A.); annalisa.monaco@univaq.it (A.M.)
2   Academy of Orofacial Myofunctional Therapy (AOMT), Los Angeles, CA 90272, USA; lcpaskay@aomtinfo.org
\*   Correspondence: sabinasaccomanno@hotmail.it

**Abstract:** Background: This study is a systematic literature review aiming at identifying the variation of the average nasolabial angle (NLA) in various orthodontic situations. The NLA is one of the key factors to be studied in an orthodontic diagnosis for the aesthetics of the nose and facial profile. Methods: Out of 3118 articles resulting from four search engines (PubMed, Cochrane Library, Turning Research Into Practice (TRIP) and SciELO), the final study allowed the analysis and comparison of only 26 studies. These included studies have considered the NLA in the following cases: teeth extraction, class II malocclusion, class III malocclusion, rapid palatal expansion (RPE), orthognathic surgery, and non-surgical rhinoplasty with a hyaluronic acid filler. Results: The results indicate that teeth extraction and the use of hyaluronic acid fillers significantly affect the NLA. Conclusions: This systematic review shows that a statistically significant change in NLA values occurs in: extractive treatments of all four of the first or second premolars in class I patients; in class II patients with upper maxillary protrusion; in patients with maxillary biprotrusion, except for cases of severe crowding; and in patients undergoing non-surgical rhinoplasty with a hyaluronic acid filler. Trial registration number: PROSPERO CRD42020185166

**Keywords:** nasolabial angle; soft-tissue profile; facial profile; orthodontic profile; rapid palatal expansion; orthognathic surgery; headgear treatment; hyaluronic acid; hyaluronic acid filler

## 1. Introduction

The evaluation of the facial profile is an important factor in any current orthodontic diagnosis, considering that an extreme advancement or retreat in the position of the upper lip or chin can determine the worsening of the patient's profile and aesthetic results. There are many factors to consider to preserve the aesthetics of the facial profile: the nasolabial angle (NLA), the nasal prominence, the position of the upper and lower lip and the depth of the chin-labial sulcus. The NLA is one of the key factors to be considered in an orthodontic diagnosis as guidance for the aesthetics of the nose and facial profile. It is defined as the angle formed by the two lines passing through the lower edge of the nose (the columella) and the edge of the upper lip (as shown in Figure 1). As described in literature, the ideal nasolabial angle ranges from 90° and 95° for males and 95–115° for females, although these values may vary among the various phenotypical groups (races).

In general, the NLA values are increased in Asian populations, showing a flatter profile and a more obtuse angle in comparison with Caucasian populations or with African populations [1]. Dental extractions in orthodontics are often recommended in order to gain space in the case of severe or moderate crowding or for the camouflage of consolidated skeletal malocclusions. However, the orthodontic diagnosis and decision of whether or not to treat a patient with extractions remains controversial. As for the NLA modification, many theories exist. Some orthodontists believe that extractions worsen the patient's soft

tissue profile by making it too flat. Therefore, they prefer to follow a conservative approach, creating minimally invasive treatments that allow the recovery of space within the arch, such as: dental stripping, expansion of the dental arches and distalization [2]. On the other hand, there are orthodontists who believe that extraction represents the right therapeutic strategy. According to this approach, extraction does not worsen the profile, indeed in some cases it may improve it, by increasing the NLA values [3]. Furthermore, extractions often have a strong positive impact on factors such as: vertical dimension, treatment stability, width of the arches, facial convexity and perioral tissues [4]. The drawback of extractions is that they may result in a bi-retruded or flat profile, narrow dental arches, or an increase in the width of buccal corridors, which, in non-extractive cases, may result in an instability of the occlusal contacts with the risk of relapse. Additionally, in borderline cases, this may lead to a biprotruded profile [5]. Therefore, the customary extraction of four premolars is contraindicated in cases of a deep bite and horizontal growth pattern, as it will lead to the loss of the vertical dimension, further retrusion of the lower third of the face and an increase in NLA values [6]. Among non-extractive protocols that may influence the NLA, there are the distalization of the upper molars and the use of mandibular advancement devices (MAD), for the treatment of class II malocclusions [7,8]. Treatments such as the Delaire face mask for maxillary advancement, orthodontic camouflage and orthognathic surgery represent valid non-extractive therapeutic options for the treatment of patients with skeletal class III [9–11]. Rapid palatal expansion (RPE) is an effective treatment option for the correction of transverse maxillary discrepancies, which produces changes in the soft tissues related to an increase in the size of the nasal soft tissues and bone bases. In this systematic review of the relevant literature, we will analyze how these various therapeutic procedures can affect the NLA values, including non-surgical rhinoplasty with a hyaluronic acid filler as a non-orthodontic treatment option to improve the patients' soft tissue profile, mostly in patients who do not want to undergo orthodontic or surgical treatment or in those cases in which the orthodontic treatment does not obtain the desired results in terms of aesthetics of the facial profile. The purpose of this systematic literature review is to study the variation of the nasolabial angle (NLA) in orthodontic therapies that include: tooth extractions; orthognathic surgery; corrections of second and third class malocclusions; rapid palate expansion (RPE) and non-surgical rhinoplasty through hyaluronic acid fillers.

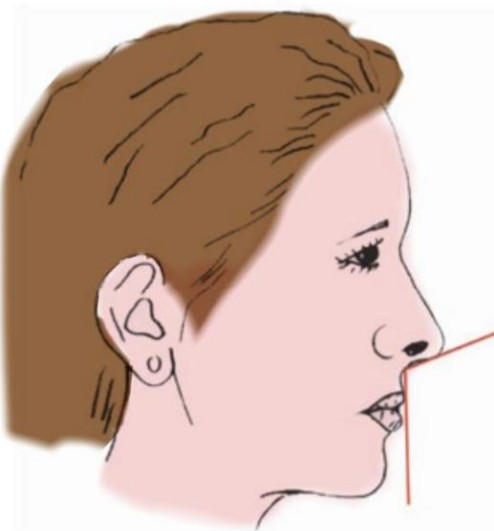

**Figure 1.** The nasolabial angle (NLA).

## 2. Materials and Methods

### 2.1. Protocol and Registration

This systematic review was conducted following the guidelines of the Cochrane Handbook for Systematic Reviews of Interventions [12]. The methods of analysis and the

inclusion criteria were specified in advance and documented in a protocol registered in the National Institute of HealthResearch database (http://www.crd.york.ac.uk/prospero/ (accessed on 15 July 2020); trial registration number: CRD42020185166).

### 2.2. Eligibility Criteria

The studies included in the present systematic review were bibliographic review, systematic review, meta-analysis analyzing the profilometric variations of the nasolabial angle in orthodontic therapies that include dental extractions, orthognathic surgery, corrections of III and II class malocclusions, rapid palatal expansion, and non-surgical rhinoplasty with a hyaluronic acid filler in healthy patients. Observational studies on untreated patients, comparative studies, clinical trials, case reports, and opinion articles were excluded. Excluded as well were studies analyzing syndromes (e.g., cleft lip), or studies not including the measurement of NLA values as part of the profilometric evaluation of the patient's soft tissues.

### 2.3. Information Sources, Search Strategy, and Study Selection

This systematic literature review was carried out by searching articles on: PubMed, Cochrane Library, Turning Research into Practice (TRIP) and SciELO. The following key words have been used: "nasolabial angle", "soft-tissue changes", "soft-tissue profile changes", "facial profile changes", "orthodontic profile changes", "soft-tissue profile hyaluronic acid filler", "headgear treatment nasolabial angle" and "nasolabial angle hyaluronic acid". Eligibility was discussed by the authors by screening the title and abstracts of the retrieved articles. Whenever in doubt about the inclusion or exclusion of an abstract, the full text was accessed. The study's screening and selection process are represented in the PRISMA flow diagram of Figure 2.

### 2.4. Data Items

The main outcomes retrieved from the articles were the measurements in degree of the NLA values of patients treated for the resolution of malocclusion or for the improvement of the facial profile through non-surgical rhinoplasty with a hyaluronic acid filler. As mentioned earlier, articles were not considered that focused on fillers but not the NLA.

### 2.5. Summary Measures and Approach to Synthesis

The studies included considered values in degrees of NLA in the following cases: teeth extraction, class II malocclusion, class III malocclusion, rapid palatal expansion (RPE), orthognathic surgery, and non-surgical rhinoplasty with a hyaluronic acid filler. The angular values of NLA are summarized in Table 1A–F, while Table 2A–F summarizes the age and sex data of the patients studied. Total number of patients, their gender, and average age are summarized in Table 3. The summary and explanation of all abbreviations used in all tables and text are summarized in Table 4.

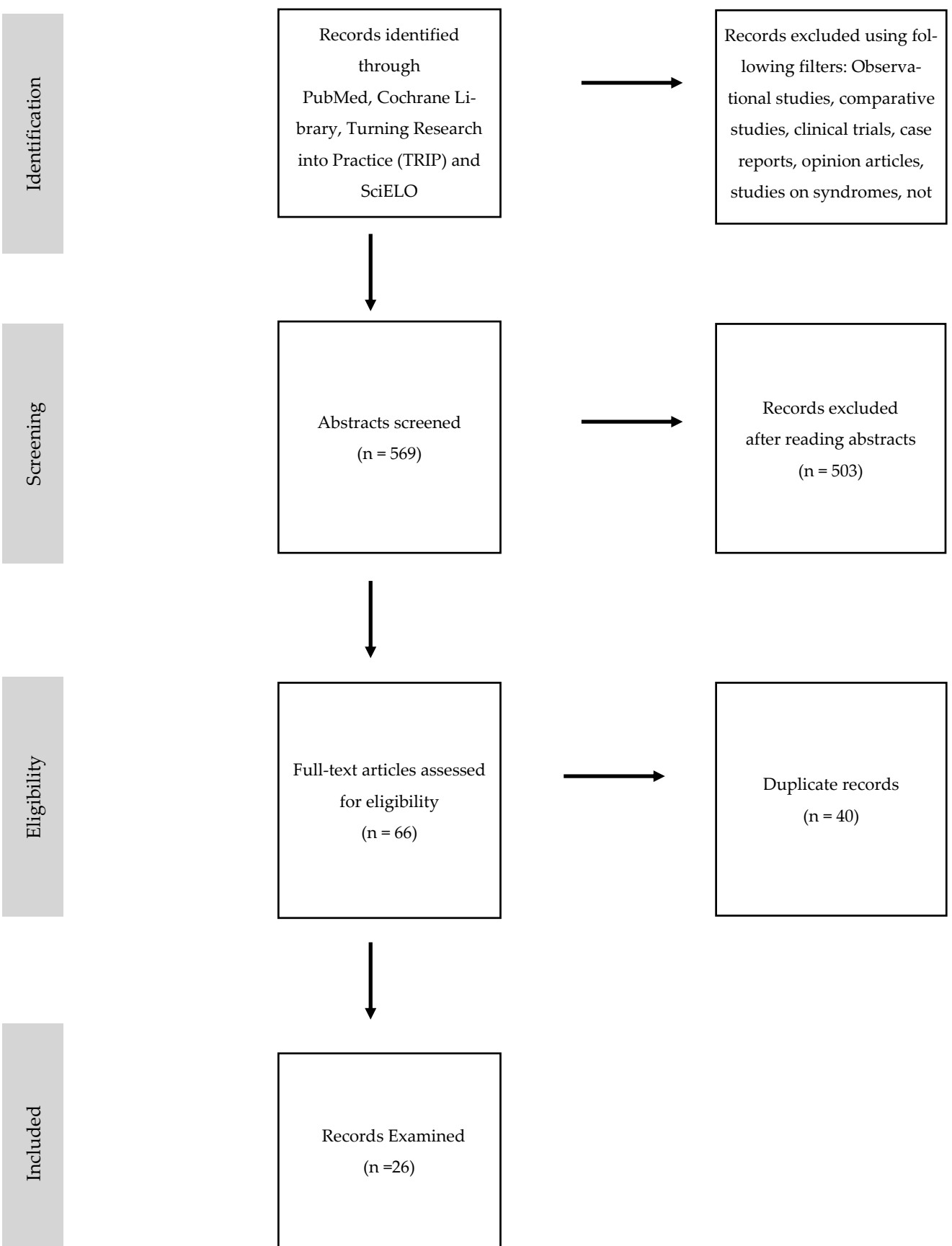

**Figure 2.** PRISMA flow diagram showing the study's screening and selection process.

**Table 1.** (**A**) Results of extractions on NLA average value; (**B**) Results of Class II malocclusion on NLA average value; (**C**) Results of Class III malocclusion on NLA average value; (**D**) Results of rapid maxillary expansion on NLA average value; (**E**) Results of orthognatic surgery on NLA average value; (**F**) Results of hyaluronic acid fillers on NLA average value.

| | Article Title | Year | Author | Result | NLA Average Value |
|---|---|---|---|---|---|
| | | | **(A)** | | |
| Extractions | Changes In Soft Tissue Profile After Orthodontic Treatment With And Without Extraction: A Systematic Review And Meta-Analysis [3] | 2018 | Rian H. Almurtadha Et Al. | NLA Increase (4.92° Average Value) In Extractive Group Compared To Non-Extractive Group | Not Reported |
| Extractions | Post-Orthodontic Cephalometric Variations In Bimaxillary Protrusion Cases Managed By Premolar Extraction—A Retrospective Study [13] | 2019 | Nd Alqahtani Et Al. | Significant NLA Increase (6.6°) In Extractive Group Compared To Non-Extractive Group | 104.7° ± 18.4° |
| Extractions | Soft Tissue Changes Following Extraction Vs. Nonextraction Orthodontic Fixed Appliance Treatment: A Systematic Review And Meta-Analysis [4] | 2018 | Konstantonios D Et Al. | Significant NLA Increase (2.4–4.3°) In Extractive Group Compared To Non-Extractive Group | Not Reported |
| Extractions | Soft Tissue Change In Patients With Dentoalveolar Protrusion Treated With Maximum Anchorage: A Systematic Review And Meta-Analysis [14] | 2019 | Yan Liu Et Al. | Higher NLA Increase Miniscrew Anchorage (3.52°) Compared To Traditional Anchorage (0.68°) | Not Reported |
| Extractions | Comparison Of Treatment Effects Between Four Premolar Extraction And Total Arch Distalization Using The Modified C-Palatal Plate [2] | 2018 | Sung Young Jo Et Al. | Significant NLA Increase In Both Study Groups | Extr. 109.21° ± 11.72° Distalization 104.53° ± 11.55° |
| Extractions | Soft Tissue Facial Profile Changes After Orthodontic Treatment With Or Without Tooth Extractions In Class I Malocclusion Patients: A Comparative Study [15] | 2019 | Benedito V. Freitas Et Al. | No Statistically Significant Variations On Average NLA Values Between Extractive And Non-Extractive Group | Extr. 106.4° No Extr. 97.1° |
| Extractions | Esthetic Perception Of Changes In Facial Profile Resulting From Orthodontic Treatment With Extraction Of Premolar [16] | 2017 | Walter Iared Et Al. | 1.4° NLA Increase In Extractive Group And 3° Decrease In Non-Extractive Group | Not Reported |
| Extractions | Comparative Evaluation Of Soft Tissue Changes In Class I Borderline Patient Treated With Extraction And Nonextraction Modalities [5] | 2016 | Aniruddh Yashwant V. Et Al. | Significant NLA Increase (9.410°) In Extractive Group Compared To Non-Extractive Group | Extr. 102.77° No Extr. 98.23° |
| Extractions | Profile Changes Following Extraction Orthodontic Treatment: A Comparison Of First Versus Second Premolar Extraction [17] | 2018 | Ziad Omar Et Al. | Significant NLA Increase (2.2°) In Extractive Group Compared To Non-Extractive Group | Pm1 112.81° ± 8.01° Pm2 113.13° ± 7.76° |
| Extractions | Short-Term Effects Of Systematic Premolar Extraction On Lip Profile, Vertical Dimension And Cephalometric Parameters In Borderline Patients For Extraction Therapy—A Retrospective Cohort Study [6] | 2016 | Christian Kirschneck Et Al. | No Statistically Significant Variations In NLA Values Between Extractive And Non-Extractive Group | Extr. 113.2° No Extr. 112.7° |

**Table 1.** *Cont.*

| | | | | | |
|---|---|---|---|---|---|
| **(B)** | | | | | |
| Class II Malocclusion | Morphological Characteristics Of Soft Tissue Profile Of Angle's Class II Division I Malocclusion Before And After Orthodontic Treatment [18] | 2018 | Jing Xuan Et Al. | Increase In NLA Values After Treatment, But Values Are Not Statistically Significant | Not Reported |
| Class II Malocclusion | Soft Tissue Profile Changes After Functional Mandibolar Advancer Or Herbst Appliances Treatment In Class II Patients [7] | 2017 | Jan Hourfar Et Al. | No Statistical Differences Were Detected Between Both Ffas And Rfa, NLA Showed More Pronounced Changes In Herbst Appliance Patients. | 114.78° Fma 118.64° Herbst |
| Class II Malocclusion | Class II Malocclusion Treatment Effects With Jones Jig And Distal Jet Followed By Fixed Appliances [8] | 2018 | Lorena Vilanova Et Al. | Better NLA Values In Control Group (Untreated Patients) | 107.25° |
| Class II Malocclusion | Effectiveness Of Early Orthopaedic Treatment With Headgear: A Systematic Review And Meta-Analysis [19] | 2017 | Spyridon N. Papageorgiou Et Al. | Short-Term Decrease In NLA Values | 81.6° |
| Class II Malocclusion | Comparison Of Treatment Effects Between The Modifies C-Palatal Plate And Cervical Pull Headgear For Total Arch Distalization In Adults [20] | 2017 | Chong Ook Park Et Al. | Significant Increase In NLA And Upper Lip Retraction | 81.31° |
| **(C)** | | | | | |
| Class III Malocclusion | Comparison Of The Soft And Hard Tissue Effects Of Two Different Protraction Mechanisms In Class III Patients: A Randomized Clinical Trial [9] | 2015 | Mevlut Celikoglu Et Al. | No Statistically Significant Difference In Both Study Groups | Rme/Fm 98.33° Mmp 106.24° |
| Class III Malocclusion | Morphological Changes Of Skeletal Class III Malocclusion In Mixed Dentition With Protraction Combined Activities [10] | 2018 | Fan-Yu Xu Et Al. | Significant NLA Decrease 5.629° | Not Reported |
| **(D)** | | | | | |
| Rapid Maxillary Expansion | Soft Tissue Changes In The Orofacial Region After Rapid Maxillary Expansion [21] | 2016 | Gulsilay Sayar Torun | No Statistically Significant Difference In Both Study Groups | 117.5° |
| **(E)** | | | | | |
| Orthognatic Surgery | Does Maxillary Advancement Influence The Nasolabial Angle? [22] | 2019 | Tom Shmuly Et Al. | No Statistically Significant Difference In Both Study Groups NLA Decrease 3.78° | 98.68° ± 12.10° |
| Orthognatic Surgery | Nasolabial Changes Following Double Jaw Surgery [23] | 2019 | Michelle L. Allar Et Al. | NLA Decrease Correlated With Maxillary And Mandibular Advancement | Not Reported |
| Orthognatic Surgery | Maxillary Advancement Versus Mandibular Setback In Class III Dentofacial Deformity:Are There Any Differences In Aesthetic Outcomes? [11] | 2016 | M. Ghassemi Et Al. | NLA Increase After Mandibular Setback NLA Decrease After Upper Maxillary Advancement | Mandibular Setback 106.585° Upper Maxillary Advanement 102.075° |
| Orthognatic Surgery | Effect Of Maxillary Advancement On The Changes In The Soft Tissues After Treatment Of Patients With Class III Malocclusion [24] | 2015 | M. Ghassemi Et Al. | No Statistically Significant Difference In Both Study Groups | 100.6° |

**Table 1.** *Cont.*

| (F) | | | | | |
|---|---|---|---|---|---|
| Hyaluronic Acid Filler | Non-Surgical Rhinoplasty With Hyaluronic Acid Fillers: Predictable Results Using Software For The Evaluation Of Nasal Angles [25] | 2020 | Adriano Santorelli Et Al. | Significant Increase In NLA | $88.5° \pm 6.1°$ |
| Hyaluronic Acid Filler | Midline Volume Filler Injection For Facial Rejuvenation And Countouring In Asians [26] | 2019 | Zhezhen Xiong Et Al. | NLA Increase $3.2° \pm 2.6°$ | $107.93° \pm 9.01°$ |
| Hyaluronic Acid Filler | Early Changes In Facial Profile Following Structured Filler Phinoplasty: An Anthropometric Analysisi Using A 3-Dimensional Imaging System [1] | 2017 | Nark Kyoung Rho Et Al. | NLA Increase $3.79° \pm 8.71°$ | $95.97°$ |
| Hyaluronic Acid Filler | Filler Rhinoplasty Evaluated By Anthropometric Analysis [27] | 2016 | Sung Hwan Youn Et Al. | NLA Increase $9.4° \pm 4.5°$ | Increase: Tip Rotation $93.3° \pm 9.3°$ Total Nose $96.9° \pm 10°$ Hump Correction $96.8° \pm 10°$ |

**Table 2.** (**A**) Number, gender and average age in extraction cases; (**B**) Number, gender and average age in Class II malocclusion case; (**C**) Number, gender and average age in Class III malocclusion cases; (**D**) Number, gender and average age in rapid maxillary expansion cases; (**E**) Number, gender and average age in orthognatic surgery cases; (**F**) Number, gender and average age in hyaluronic acid filler cases.

| (A) | | | |
|---|---|---|---|
| | **Article Title** | **Number and Age of Reported Patients** | **Gender** | **Average Age** |
|---|---|---|---|---|
| Extractions | Changes In Soft Tissue Profile After Orthodontic Treatment With And Without Extraction: A Systematic Review And Meta-Analysis [3] | 305 Patients 12.26–24.49 Years Old | Extr 102f 18m No Extr. 77f 16m Not Reported: 92 Patients | 18.375 Years Old |
| Extractions | Post-Orthodontic Cephalometric Variations In Bimaxillary Protrusion Cases Managed By Premolar Extraction—A Retrospective Study [13] | 46 Patients 18–30 Years Old | 30f 16m | 24 Years Old |
| Extractions | Soft Tissue Changes Following Extraction Vs. Non Extraction Orthodontic Fixed Appliance Treatment: A Systematic Review And Meta-Analysis [4] | 1876 Patients. 14 Years Old | 1291f 585m | 14 Years Old |
| Extractions | Soft Tissue Change In Patients With Dentoalveolar Protrusion Treated With Maximum Anchorage: A Systematic Review And Meta-Analysis [14] | 99 Patients. 13.55–29.25 Years Old | Not Reported | 21.4 Years Old |
| Extractions | Comparison Of Treatment Effects Between Four Premolar Extraction And Total Arch Distalization Using The Modified C-Palatal Plate [2] | 40 Patients 16.1–32.2 Years Old | 4pe 19f 1m Mcpp 16f 4m | 24.15 Years Old |

**Table 2.** *Cont.*

| (A) | | | | |
|---|---|---|---|---|
| | **Article Title** | **Number and Age of Reported Patients** | **Gender** | **Average Age** |
| Extractions | Soft Tissue Facial Profile Changes After Orthodontic Treatment With Or Without Tooth Extractions In Class I Malocclusion Patients: A Comparative Study [15] | 20 Patients. 12.3 Years Old | Extr. 6f 4m No Extr. 5f 5m | 12.3 Years Old |
| Extractions | Esthetic Perception Of Changes In Facial Profile Resulting From Orthodontic Treatment With Extraction Of Premolar [16] | 195 Patients Average Age Not Reported | Not Reported | Not Reported |
| Extractions | Comparative Evaluation Of Soft Tissue Changes In Class I Borderline Patient Treated With Extraction And Non Extraction Modalities [5] | 150 Patients Average Age Not Reported | Not Reported | Not Reported |
| Extractions | Profile Changes Following Extraction Orthodontic Treatment: A Comparison Of First Versus Second Premolar Extraction [17] | 81 Patients 10–16 Years Old | Pm1 28f 20m Pm2 22f 11m | 13 Years Old |
| Extractions | Short-Term Effects Of Systematic Premolar Extraction On Lip Profile, Vertical Dimension And Cephalometric Parameters In Borderline Patients For Extraction Therapy—A Retrospective Cohort Study [6] | 50 Patients 9–15 Years Old | Extr. 15f 10m No Extr. 14f 11m | 12 Years Old |
| | 345 Patients Not Included In The Calculations Of Average Values, Due To Incomplete Data (Patients' Average Age) Provided By The Authors | Total Average Age Total Patients Reported 2517 | | 15.07 Years Old |
| (B) | | | | |
| II Class Malocclusion | Morphological Characteristics Of Soft Tissue Profile Of Angle's Class II Division I Malocclusion Before And After Orthodontic Treatment [18] | 20 Patients 14.3 Years Old | Not Reported | 14.3 Years Old |
| II Class Malocclusion | Soft Tissue Profile Changes After Functional Mandibular Advancer Or Herbst Appliances Treatment In Class II Patients [7] | 42 Patients 12.1–16.2 Years Old | Fma 10f 11m Herbst 10f 11m | 14.15 Years Old |
| II Class Malocclusion | Class II Malocclusion Treatment Effects With Jones Jig And Distal Jet Followed By Fixed Appliances [8] | 45 Patients 12.90 Years Old | 16f 29m | 12.9 Years Old |
| II Class Malocclusion | Effectiveness Of Early Orthopaedic Treatment With Headgear: A Systematic Review And Meta-Analysis [19] | 930 Patients 7.6–12.9 Years Old | 379f 479m | 10.25 Years Old |
| II Class Malocclusion | Comparison Of Treatment Effects Between The Modifies C-Palatal Plate And Cervical Pull Headgear For Total Arch Distalization In Adults [20] | 44 Patients. Mcpp 24.7 ± 7.7 Years Old Headgear 23.0 ± 7.7 Years Old | 32f 12m | 23.85 Years Old |
| | | Total Average Age Total Patients Reported 1081 | | 11.14 Years Old |

**Table 2.** *Cont.*

| | (C) | | | |
|---|---|---|---|---|
| III Class Malocclusion | Comparison Of The Soft And Hard Tissue Effects Of Two Different Protraction Mechanisms In Class III Patients: A Randomized Clinical Trial [9] | 32 Patients 12 ± 0.89 Years Old | Mmp 9f 7m Fm/Rme 10f 6m | 12 Years Old |
| III Class Malocclusion | Morphological Changes Of Skeletal Class III Malocclusion In Mixed Dentition With Protraction Combined Activities [10] | 30 Patients 6–10 Years Old | 15f 15m | 8 Years Old |
| | | Total Average Age Total Patients Reported 62 | | 10.6 Years Old |
| | (D) | | | |
| Rapid Maxillary Expansion | Soft Tissue Changes In The Orofacial Region After Rapid Maxillary Expansion [21] | 28 Patients. 13.91 ± 1.8 Years Old | 18f 10m | 13.91 Years Old |
| | | Total Average Age Total Patients Reported 28 | | 13.91 Years Old |
| | (E) | | | |
| Orthognatic Surgery | Does Maxillary Advancement Influence The Nasolabial Angle? [22] | 32 Patients 21 ± 3.4 Years Old | 20f 12m | 21 Years Old |
| Orthognatic Surgery | Nasolabial Changes Following Double Jaw Surgery [23] | 37 Patients 32.2 ± 14.6 Years Old | 29f 8m | 32.2 Years Old |
| Orthognatic Surgery | Maxillary Advancement Versus Mandibular Setback In Class III Dentofacial Deformity: Are There Any Differences In Aesthetic Outcomes? [11] | 34 Patients 16–51 Years Old | Not Reported | 33.5 Years Old |
| Orthognatic Surgery | Effect Of Maxillary Advancement On The Changes In The Soft Tissues After Treatment Of Patients With Class III Malocclusion [24] | 48 Patients 28 Years Old | 29f 24m | 28 Years Old |
| | | Total Average Age Total Patients Reported 151 | | 28.78 Years Old |
| | (F) | | | |
| Hyaluronic Acid Filler | Non-Surgical Rhinoplasty With Hyaluronic Acid Fillers: Predictable Results Using Software For The Evaluation Of Nasal Angles [25] | 62 Patients 29 ± 9.2 Years Old | 57f 5m | 29 Years Old |
| Hyaluronic Acid Filler | Midline Volume Filler Injection For Facial Rejuvenation And Countouring In Asians [26] | 40 Patients 31.55 ± 6.43 Years Old | 37f 3m | 31.55 Years Old |
| Hyaluronic Acid Filler | Early Changes In Facial Profile Following Structured Filler Rhinoplasty: An Anthropometric Analysis Using A 3-Dimensional Imaging System [1] | 40 Patients 28.5 Years Old | 40f 0m | 28.5 Years Old |
| Hyaluronic Acid Filler | Filler Rhinoplasty Evaluated By Anthropometric Analysis [27] | 242 Patients 31 ± 9 Years Old | Not Reported | 31 Years Old |
| | | Total Average Age Total Patients Reported 384 | | 30.47 Years Old |

**Table 3.** Summary of total number of patients, gender, and average age.

| Total Number of Patients Reported | Total Number of Patients by Gender | Total Average Age | Not Classified by Gender | Not Classified by Age |
|---|---|---|---|---|
| 4568 | 2336 F<br>1333 M | 14.62 Years old | 897 | 345 |

**Table 4.** Abbreviations.

| Abbreviation | Meaning |
|---|---|
| NLA | Nasolabial Angle |
| Extr. | Patients Treated With Extraction |
| No Extr. | Patients Treated Without Extraction |
| Pm1 | Patients Treated With Extraction Of First Four Premolars |
| Pm2 | Patients Treated With Extraction Of Second Four Premolars |
| Rfa | Patients Treated With Removable Functional Appliances |
| Herbst | Patients Treated With Herbst Appliance |
| Ffas | Patients Treated With Fixed Functional Appliances |
| Fma | Patients Treated With Functional Mandibular Advancer |
| Rme/Fm | Face Mask And Rapid Maxillary Expansion |
| Mmp | Mini Maxillary Protractor |
| F | Female Patients |
| M | Male Patients |
| 4pe | Four Premolar Extraction |
| Mcpp | Total Arch Distalization Using The Modified C-Palatal Plate |

## 3. Results

The bibliographic research for this systematic literature review led to the selection of 26 articles that dealt with the profilometric variations of the soft tissues that occur in orthodontic patients treated according to the malocclusion presented, and in patients undergoing non-surgical rhinoplasty with a hyaluronic acid filler. Particular attention was paid to the changes in the nasolabial angle (NLA) found in the different study groups. All patients were considered otherwise healthy if they had no medical or developmental conditions which would exclude them from the studies by the authors of the various articles. Although most of the time the gender of the patients was mentioned, their race and age were infrequently specified, therefore these two variables were not included in the analysis nor in various comparisons between subject groups, except for one study [4] that compared the impact of age on the NLA variation values. As gender was not always mentioned and most studies had mixed gender groups, it was not possible to determine the difference in NLA increase or decrease with various therapeutic modalities in males vs. females. Out of the 26 selected articles:

- 10 articles focused on NLA in extractive cases.

The authors analyzed the changes in the NLA recorded after the treatment of 2862 patients. Four authors compared the changes in the NLA as recorded in the group of patients who underwent extractions of two or four premolars (*patients with extraction*) and in the group of patients on whom no extractions were performed (*non-extractive patients*), registering a significant increase in the NLA, which varies from 2.4° to 9.410° in extractive post-treatment patients. The four authors compared the NLA variations on 346 patients overall, and found no statistically significant differences between the groups of patients studied. One author compared the changes in the NLA that occur in extractive patients treated with skeletal anchorage by mini screws and in patients treated with traditional anchorage, registering a greater reduction in the NLA values in the first group (3.52°). Conversely, another author studied the NLA variations by comparing the extractive group to a group of patients on which distalization with a C-palatal plate (non-extractive therapy) was performed, registering a significant increase in the NLA in both study groups.

- 5 articles analyzed the NLA in cases of class II malocclusion.

In three studies, the authors analyzed the variations in the NLA recorded in 107 patients with class II malocclusion. There were no statistically significant NLA values after orthodontic treatment in patients with class II malocclusions treated with Jones Jig and Distal Jet followed by fixed appliances, Herbst or mandibular advancer among the three studies' groups. In the two remaining studies, 974 patients with class II malocclusions were treated with extra-oral tractions. The evaluation of the NLA values, one year after headgear treatment, showed a reduction of that value on 930 patients, while a significant increase in the NLA and upper lip retraction was achieved on 44 patients.

- 2 articles studied the NLA in class III malocclusion cases.

The authors studied changes in the NLA recorded in 62 patients with class III malocclusion. One author recorded a significant decrease in the NLA values in patients treated with maxillary protrusion devices, while the other study found no statistically significant NLA values after treatment with different maxillary protrusion systems.

- 1 article focused on the NLA in cases of rapid palatal expansion.

In this study, conducted on 28 patients, the authors did not record statistically significant NLA values after treatment with a rapid palate expander.

- 4 articles analyzed the NLA in cases of orthognathic surgery.

There were no statistically significant changes in these studies conducted on a total of 151 surgical patients undergoing maxillary advancement.

- 4 articles studied the variation of the NLA in relation to the treatment of non-surgical rhinoplasty with a hyaluronic acid filler.

The authors studied the variations of the NLA recorded after the treatment of a total of 384 patients with a hyaluronic acid filler. The four studies showed a statistically significant increase in NLA values which reached 13.9°.

## 4. Discussion

### 4.1. NLA in Extraction vs. Non-Extraction Cases

There were 10 articles addressing the variations of the NLA by comparing the extractive cases to the non-extractive cases, with a total of 2845 patients divided into the various study groups, including 428 females and 696 males, with an average age of 14.62 years. Among these, four authors found a statistically significant increase in post-treatment NLA values ranging from 2.4° to 9.410° in class I patients treated with extractions of the first four or second premolars, and in patients with maxillary bi-protrusion treated with extractions of all four first premolars. Almurtadha et al. recorded a retraction in the position of the upper and lower lips and the consequent increase in the NLA with an average difference of 4.92° after the orthodontic treatment of these patients who had extractions. The authors explained that the retraction of the lips, and therefore an increase of the NLA, is due to the retraction of the upper and lower anterior teeth as they move backwards to occupy the extractive spaces. On the other hand, there is no change in the NLA in those cases with extractive orthodontics where the procedure indicated aimed at the resolution of dentoalveolar crowding or a camouflage of mild skeletal malocclusions. The authors stated that a 2 mm change in lip retraction is enough to make the profile look worse or better [3]. Similar results to the previous study were reported by Alqahtani et al., who registered a statistically significant increase both in the NLA and in the length of the upper lip, as related to a retraction of the upper and lower incisors—plus a retro-inclination—in biprotruded patients treated with extractions of the first upper and lower premolars. They also report a decrease of 1.1 mm post treatment in the exposure of the upper incisors [13]. Additionally, Konstantonios et al. reported a significant increase in the NLA in cases involving the extraction of the four upper and lower first premolars compared to non-extractive cases. However, there was no significant increase in the NLA's values obtained in cases where

only two premolars were extracted. The retraction of the lips and the increase in the NLA's values were associated with a retraction of the upper or lower incisors. This means that extractions can have a low impact on the facial profile if the retraction of the anterior sector is minimal, as it occurs in cases where the extractive spaces are closed by mesialization of the posterior sectors.

There was also a significant association between the NLA values and the patients' age, as a greater increase in the NLA was achieved in older patients than in those youngsters who were still growing. The cause of this association is reported to be related to the growth of the nasal spine or of the lips during the patient's growth [4]. A significant increase in NLA of 9.410°—with subsequent labial retraction in class I borderline patients—was recorded in the study of Yashwant, Ravi and Arumugam, which supports the results reported by the previous authors [5].

Four studies reported no statistically significant values for the NLA by comparing the results obtained among a group of patients undergoing extractions of the four premolars and a non-extractive group, among these: Freitas et al. in their study on 20 class I patients, Iared et al. in their study conducted on 195 patients, and Kirschneck et al. in their study on 50 patients [6,15,16]. Omar et al., as well, registered no statistically significant NLA values, but in their article they compared the NLA values obtained in two extraction groups, one with the extraction of the four first premolars and one with the extraction of four s premolars [17].

Two more studies reported a statistically significant increase of NLA values in the treatment of extractive cases. One of these studied the results obtained with skeletal anchorage using orthodontic mini screws. Liu et al. obtained a 3.52° increase in NLA values on biprotruded patients [14]. Conversely, Jo et al. in their scientific article studied variations in the NLA between a group undergoing extractions and a group undergoing molar distalization and they reported a significant increase in the values of the NLA in both study groups. Furthermore, they recommended the extraction of the four premolars in patients with class II malocclusion who required further retraction of the anterior sectors and improvement of the soft tissues profile [2].

### 4.2. NLA in Treatment of Class II Malocclusion

Regarding the variations in the NLA as recorded in studies concerning class II malocclusions, there were no statistically significant differences when patients were treated with functional orthodontic, orthopedic devices or with brackets. Xuan et al. reported a correlation between changes in the NLA and the posterior occlusal plane in patients with class II malocclusion [18]. Hourfar et al. recorded a greater increase in the NLA in patients treated with Herbst (118.64°) compared to patients treated with removable appliances for mandibular advancement (114.78°) [7]. Vilanova et al. did not find significant variations in NLA values in their study comparing the effects of two different devices (Distal Jet and Jones Jig, respectively) used for the treatment of class II malocclusion, with a control group of untreated patients [8].

The results for two articles evaluating the variations of the NLA obtained after treatment with extraoral tractions were contradictory. Papageorgiou et al. did not record statistically significant changes in NLA values after the "short-term" (as defined by the authors) treatment of 930 patients with extraoral tractions. The effects achieved on the NLA were associated with the degree of protrusion of the maxilla before receiving orthodontic treatment and there was no relationship with the modification of the longitudinal growth recorded in these patients. Moreover, there was no statistically significant difference between the three types of extraoral tractions studied, which were, respectively, high traction, cervical traction, and combined traction [19]. However, Park et al. did record differences that were statistically significant in their study of 44 subjects comparing patients with class II malocclusions treated with extraoral cervical traction to patients treated with C-palatal plane on orthodontic mini screws [20]. Although the results obtained by these two studies were contradictory, it should be noted that 930 patients were included in the study by

Papageorgiou et al., a much larger sample of orthodontic patients with class II malocclusion than the 44 patients treated by Park et al. Furthermore, it should be considered that the first study was about growing patients, while the second one dealt with adult patients.

### 4.3. NLA in Treatment of Class III Malocclusion

In this systematic literature review, only two articles evaluated the effects produced on the NLA by orthodontic treatment protocols for class III malocclusion. Xu et al. reported a statistically significant decrease in the NLA values of 5.629° in patients treated with protocols that included a protraction of the maxilla. They also reported changes in the position of the upper and lower lip [10]. Conversely, Celikoglu et al. reported no statistically significant difference between two groups of patients treated with a "mini maxillary protractor" and with rapid palatal expansion plus face mask. The slight increase in NLA values recorded in patients treated with "mini maxillary protractor" was certainly due to the dental and skeletal effects on the upper jaw, resulting in a better support to the upper lip, which was more advanced [9].

### 4.4. NLA after Rapid Palatal Expansion

Torun is the only author who dealt with the effects of rapid palatal expansion, reporting a slight and not statistically significant decrease in NLA values [21].

### 4.5. NLA after Orthognatic Surgery

No statistically significant changes in NLA values were reported in the four studies on orthognathic surgery, all performed by Lefort I. Shmuly et al., which found a decrease of 3.78° in the NLA in their study of variation in the NLA, in the upper lip and in the nasal base in patients who underwent upper jaw advancement surgery. However, the authors were unable to find a correlation between a decrease in the NLA and the amount of maxillary advancement [22]. Allar et al. also found a decrease in the NLA and an advancement of the upper lip in patients undergoing maxillary advancement surgery of $\geq 6$ mm, but not enough to be statistically significant [23]. The results obtained from the study by Ghassemi et al. on mandibular advancement were in line with the other authors [24]. Furthermore, the same authors (Ghassemi et al.), in their study on orthognathic surgery, compared the effects obtained on soft tissues after surgical treatment with mandibular repositioning and advancement of the maxilla, recording an increase in the NLA in both groups. In the case of mandibular advancement, it was due to the change in tension of the upper lip, while in the cases of mandibular advancement, it was due to the decrease in nasal prominence. Despite the data reported, the values recorded did not have a statistical significance [11].

### 4.6. NLA and Hyaluronic Acid Fillers

As for the variation in the NLA, after non-surgical rhinoplasty treatment with a hyaluronic acid filler, four articles were included in this systematic review of the literature, and they all showed statistically significant results in increasing the NLA. These medical aesthetic operations are requested by patients who show a depressed section in the middle third of the face, due to the downward rotation of the nasal tip that reduces the NLA [26] For this reason, filler injections to modify the NLA are applied in the area of the columella and of the nasal spine on the nasolabial joint, so as to modify the nasal tip [27]. In all the studies included here, a statistically significant increase in the NLA values was obtained. Santorelli et al. reported that post-treatment, the NLA average values were 88.5° with an average increase of 3.2° $\pm$ 2.6° [25]. Xiong et al. reported post-treatment NLA average values of 107.93° [26]. Youn et al. compared patients who underwent a total increase of nose proportions to patients who underwent filler injections only in the bridge of the nose. As a result, they reported that post-treatment, the NLA average value was 96.8° with an average increase of 9.4° $\pm$ 4.5° [27]. Rho et al. reported average post-treatment NLA values of 95° with an average increase of 3.79° $\pm$ 8.71° [1]. The statistically significant results obtained by all the authors who treated the increase of the NLA with filler allow us to

recognize the validity of the technique that can be used as a post orthodontic treatment to increase the NLA by altering the nasal tip.

## 5. Conclusions

Although variations in NLA were found in patients undergoing rapid palatal expansion, orthognathic surgery and correction of class II and III malocclusions by non-extractive protocols, these results were not statistically significant. However, a statistically significant change in NLA values occurred in patients:

- with a class I and extractive treatments of the first or second four premolars,
- with a class II and maxillary protrusion,
- with maxillary biprotrusion, except in cases of severe crowding where the extraction spaces are not used for retraction of the anterior sector,
- undergoing non-surgical rhinoplasty with a hyaluronic acid filler

The results of this systematic review are summarized in Table 5 and can assist the decision-making of the dental professionals regarding the esthetic outcome of various therapeutic methods, including a possible refining intervention with hyaluronic acid fillers. Another lesson learned from this systematic review is that the current medical literature does not provide significant guidance regarding the effects of race, gender and age on the modifications of the NLA, although this biometric data is commonly gathered during the evaluation. Identifying these characteristics across gender, age and race would guide the treatment options and assist the professional in the decision regarding the most appropriate therapeutic solution. The NLA is more than just an esthetic characteristic, as it also reflects the structural dimensions of the palate and the nose, which are implicated in multiple other functions, including breathing. However, it does determine a pleasant profile which is valued by patients usually more than by the dental professionals and therefore identifying these differences could mean happier patients and more successful orthodontic or othognatic treatments.

**Table 5.** Summary of results.

|  | Article Title | Year | Author | NLA Variations |
|---|---|---|---|---|
| Extractions | Changes In Soft Tissue Profile After Orthodontic Treatment With And Without Extraction: A Systematic Review And Meta-Analysis [3] | 2018 | Rian H. Almurtadha Et Al. | Increase With Extractions |
| Extractions | Post-Orthodontic Cephalometric Variations In Bimaxillary Protrusion Cases Managed By Premolar Extraction—A Retrospective Study [13] | 2019 | Nd Alqahtani Et Al. | Increase With Extractions |
| Extractions | Soft Tissue Changes Following Extraction Vs. Nonextraction Orthodontic Fixed Appliance Treatment: A Systematic Review And Meta-Analysis [4] | 2018 | Konstantonios D Et Al. | Increase With Extraction |
| Extractions | Soft Tissue Change In Patients With Dentoalveolar Protrusion Treated With Maximum Anchorage: A Systematic Review And Meta-Analysis [14] | 2019 | Yan Liu Et Al. | Increase With Extraction And Miniscrew Anchorage |
| Extractions | Comparison Of Treatment Effects Between Four Premolar Extraction And Total Arch Distalization Using The Modified C-Palatal Plate [2] | 2018 | Sung Young Jo Et Al. | Increase With Extractions |

**Table 5.** *Cont.*

| | Article Title | Year | Author | NLA Variations |
|---|---|---|---|---|
| Extractions | Soft Tissue Facial Profile Changes After Orthodontic Treatment With Or Without Tooth Extractions In Class I Malocclusion Patients: A Comparative Study [15] | 2019 | Benedito V. Freitas Et Al. | Unaffected |
| Extractions | Esthetic Perception Of Changes In Facial Profile Resulting From Orthodontic Treatment With Extraction Of Premolar [16] | 2017 | Walter Iared Et Al. | Increase With Extractions |
| Extractions | Comparative Evaluation Of Soft Tissue Changes In Class I Borderline Patient Treated With Extraction And Nonextraction Modalities [5] | 2016 | Aniruddh Yashwant V. Et Al. | Increase With Extractions |
| Extractions | Profile Changes Following Extraction Orthodontic Treatment: A Comparison Of First Versus Second Premolar Extraction [17] | 2018 | Ziad Omar Et Al. | Increase With Extractions |
| Extractions | Short-Term Effects Of Systematic Premolar Extraction On Lip Profile, Vertical Dimension And Cephalometric Parameters In Borderline Patients For Extraction Therapy—A Retrospective Cohort Study [6] | 2016 | Christian Kirschneck Et Al. | Unaffected |
| Class II Malocclusion | Morphological Characteristics Of Soft Tissue Profile Of Angle's Class II Division I Malocclusion Before And After Orthodontic Treatment [18] | 2018 | Jing Xuan Et Al. | Increase After Treatment |
| Class II Malocclusion | Soft Tissue Profile Changes After Functional Mandibolar Advancer Or Herbst Appliances Treatment In Class II Patients [7] | 2017 | Jan Hourfar Et Al. | Unaffected |
| Class II Malocclusion | Class II Malocclusion Treatment Effects With Jones Jig And Distal Jet Followed By Fixed Appliances [8] | 2018 | Lorena Vilanova Et Al. | Unaffected |
| Class II Malocclusion | Effectiveness Of Early Orthopaedic Treatment With Headgear: A Systematic Review And Meta-Analysis [19] | 2017 | Spyridon N. Papageorgiou Et Al. | Decrease After Treatment |
| Class II Malocclusion | Comparison Of Treatment Effects Between The Modifies C-Palatal Plate And Cervical Pull Headgear For Total Arch Distalization In Adults [20] | 2017 | Chong Ook Park Et Al. | Increase After Treatment |
| Class III Malocclusion | Comparison Of The Soft And Hard Tissue Effects Of Two Different Protraction Mechanisms In Class III Patients: A Randomized Clinical Trial [9] | 2015 | Mevlut Celikoglu Et Al. | Unaffected |

**Table 5.** *Cont.*

| | Article Title | Year | Author | NLA Variations |
|---|---|---|---|---|
| Class III Malocclusion | Morphological Changes Of Skeletal Class III Malocclusion In Mixed Dentition With Protraction Combined Activities [10] | 2018 | Fan-Yu Xu Et Al. | Decrease After Treatment |
| Rapid Maxillary Expansion | Soft Tissue Changes In The Orofacial Region After Rapid Maxillary Expansion [21] | 2016 | Gulsilay Sayar Torun | Unaffected |
| Orthognatic Surgery | Does Maxillary Advancement Influence The Nasolabial Angle? [22] | 2019 | Tom Shmuly Et Al. | Decrease After Treatment |
| Orthognatic Surgery | Nasolabial Changes Following Double Jaw Surgery [23] | 2019 | Michelle L. Allar Et Al. | Decrease After Treatment |
| Orthognatic Surgery | Maxillary Advancement Versus Mandibular Setback In Class III Dentofacial Deformity:Are There Any Differences In Aesthetic Outcomes? [11] | 2016 | M. Ghassemi Et Al. | Increase After Mandibular Setback Decrease After Maxillary Advancement |
| Orthognatic Surgery | Effect Of Maxillary Advancement On The Changes In The Soft Tissues After Treatment Of Patients With Class III Malocclusion [24] | 2015 | M. Ghassemi Et Al. | Unaffected |
| Hyaluronic Acid Filler | Non-Surgical Rhinoplasty With Hyaluronic Acid Fillers: Predictable Results Using Software For The Evaluation Of Nasal Angles [25] | 2020 | Adriano Santorelli Et Al. | Increase After Filler Injection |
| Hyaluronic Acid Filler | Midline Volume Filler Injection For Facial Rejuvenation And Countouring In Asians [26] | 2019 | Zhezhen Xion Et Al. | Increase After Filler Injection |
| Hyaluronic Acid Filler | Early Changes In Facial Profile Following Structured Filler Phinoplasty: An Anthropometric Analysisi Using A 3-Dimensional Imaging System [1] | 2017 | Nark Kyoung Rho Et Al. | Increase After Filler Injection |
| Hyaluronic Acid Filler | Filler Rhinoplasty Evaluated By Anthropometric Analysis [27] | 2016 | Sung Hwan Youn Et Al. | Increase After Filler Injection |

**Author Contributions:** S.S., V.Q. and A.M. were the principal investigators, N.D. and A.A. contributed in writing the manuscript and L.C.P. contributed in both editing and writing. All authors have read and agreed to the published version of the manuscript.

**Funding:** This research received no external funding.

**Institutional Review Board Statement:** Not applicable

**Informed Consent Statement:** Not applicable

**Data Availability Statement:** Not applicable

**Conflicts of Interest:** The authors declare that they have no conflict of interest.

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
