# Peer review of "Evaluation of the Nasolabial Angle in Orthodontic Diagnosis: A Systematic Review"

_applsci, doi:10.3390/app11062531_

Round 1

Reviewer 1 Report

  1. The authors should review carefully the manuscript and its conformity to the Journals' requests. In text, 'sex' should be noted as 'gender'. All abbreviations should be identified and explained (in text either at the end of the tables).
  2. Table 2 - there s a missing article title. Should be revised.
  3. All the tables should follow the same pattern - Table 3 should be revised.
  4. In the Discussion section, the authors should approach other possible and frequent factors which could contribute to NLA value modification. If there are associated disease in the patients' medical historical, it should be discussed and compared to literature data. Also, maxillary bone development should be involved and approached. 
  5. Are there in the literature studies which assessed filler treatment post orthodontic treatment as an adjuvant aesthetic therapy? If yes, they should be included in the study and discussed. 

Author Response

The reviewer comments are in blue, while the author’s responses are in red.

The current systemic review trying to summary the nasiolabial angle change caused by orthodontic treatment, which is an interested topic. However, the way the authors putting the results together causes more confusion than clarification.

  1. The authors should review carefully the manuscript and its conformity to the Journals' requests. In text, 'sex' should be noted as 'gender'. All abbreviations should be identified and explained (in text either at the end of the tables). We replaced sex with gender and we added a table of abbreviations that reflect the exact acronyms as presented in the various articles.
  2. Table 2 - there s a missing article title. Should be revised. Probably it was a formatting glitch but we fixed it
  3. All the tables should follow the same pattern - Table 3 should be revised. Done
  4. In the Discussion section, the authors should approach other possible and frequent factors which could contribute to NLA value modification. If there are associated disease in the patients' medical historical, it should be discussed and compared to literature data. Also, maxillary bone development should be involved and approached. We addressed this issue on now page 6, under Results.
  5. Are there in the literature studies which assessed filler treatment post orthodontic treatment as an adjuvant aesthetic therapy? If yes, they should be included in the study and discussed. We addressed this issue on page 4, in the Data Items section.

Thank you for your guidance and useful comments.

Reviewer 2 Report

The current systemic review trying to summary the nasiolabial angle change caused by orthodontic treatment, which is an interested topic. However, the way the authors putting the results together causes more confusion than clarification.

  1. Page 2 “According to this approach, extraction does not worsen the profile, indeed in some cases it may improve it, by reducing the NLA values.[4]”. Does the author mean “by increasing the NLA values”?
  2. Figure 1: why “systemic review, meta-analysis” are listed as inclusive criteria? The authors aimed to do meta-analysis on previous published meta-analysis instead of including original research articles?
  3. Page 5: “, one author studied the NLA variations by comparing the extractive group to a group of patients on which distalization is performed, registering a significant increase in the NLA in both study groups”. Is there any difference about the amount of NLA increasement between groups? And for the distalization group, was extract performed?
  4. Page 6: “There were no statistically significant NLA values” should be “There were no statistically significant NLA value changes”
  5. Page 6: it’s hard to follow the authors’ logic here “. There were no statistically significant NLA values after orthodontic treatment among the three studies’ groups, while in the two remaining studies, 974 patients with class II malocclusions were treated with extra-oral tractions. A short-term reduction of the NLA values was achieved on 930 patients, while a significant increase in the NLA and upper lip retraction was achieved on 44 patients”. Does the author meant there are significant NLA angle for the two studies in which 974 patients were treated with extra-oral tractions, and among these 974, 930 have NLA reduction and 44 have NLA increase? And what does “shortterm” mean? Does it mean the NLA angle went back to normal later for these 974 patients, or the study observation period is short? And for the three studies in which NLA value was not altered by orthodontic treatment, what orthodontic strategy was used? Extraction? Surgery? Independent alignment?
  6. For the studies about NLA change after orthognathic surgery, what type of the surgeries were included?
  7. Table 1 B: what does 0.629 +/- 4.652 mean? NLA angle change in degree?
  8. Table 1C: “MMP 106.24”, the symbol of degree is missing here.
  9. Table 1E, line 3, what’s the results about the NLA change for this study?
  10. The authors considered sex and age for each study, but the information of sex and age are listed separately with the NLA changes. And they are not discussed in the result section. Do the authors observed any differences on NLA change between genders or among different age groups?
  11. What about the race distribution? Different population have different thickness of the soft tissue, so their upper lip response differently to same treatment strategy.
  12. Page 15 discussion section “Authors should discuss the results and how they can be interpreted from the perspective of previous studies and of the working hypotheses. The findings and their implications should be discussed in the broadest context possible. Future research directions may also be highlighted.” These sentences are from MDPI template instruction. Same on page 17, the first sentence for conclusion section
  13. The discussion section is a whole long paragraph which repeats the research findings from other studied. It is reader unfriendly and it has no authors true discussion points.

Author Response

The reviewer comments are in blue, while the author’s responses are in red.

The current systemic review trying to summary the nasiolabial angle change caused by orthodontic treatment, which is an interested topic. However, the way the authors putting the results together causes more confusion than clarification.

  1. Page 2 “According to this approach, extraction does not worsen the profile, indeed in some cases it may improve it, by reducing the NLA values.[4]”. Does the author mean “by increasing the NLA values”? Made correction (now on page 3)
  2. Figure 1: why “systemic review, meta-analysis” are listed as inclusive criteria? The authors aimed to do meta-analysis on previous published meta-analysis instead of including original research articles? Made corrections on Flowchart of Fig. 1 (now page 5)
  3. Page 5: “, one author studied the NLA variations by comparing the extractive group to a group of patients on which distalization is performed, registering a significant increase in the NLA in both study groups”. Is there any difference about the amount of NLA increasement between groups? And for the distalization group, was extract performed? Corrections made (now page 6)
  4. Page 6: “There were no statistically significant NLA values” should be “There were no statistically significant NLA value changes” Correction made (now page 6)
  5. Page 6: it’s hard to follow the authors’ logic here “. There were no statistically significant NLA values after orthodontic treatment among the three studies’ groups, while in the two remaining studies, 974 patients with class II malocclusions were treated with extra-oral tractions. A short-term reduction of the NLA values was achieved on 930 patients, while a significant increase in the NLA and upper lip retraction was achieved on 44 patients”. We reformatted the Discussion section to make it more reader friendly. Does the author meant there are significant NLA angle for the two studies in which 974 patients were treated with extra-oral tractions, and among these 974, 930 have NLA reduction and 44 have NLA increase? Yes, that’s why we mentioned that the results were somehow contradictory. And what does “shortterm” mean? That was defined by the authors, and they later referred to a one year follow-up. Does it mean the NLA angle went back to normal later for these 974 patients, or the study observation period is short? And for the three studies in which NLA value was not altered by orthodontic treatment, what orthodontic strategy was used? Extraction? Surgery? Independent alignment? We made changes to this section realizing that an article did not qualify under further scrutiny. (now pages 16 and 17)
  6. For the studies about NLA change after orthognathic surgery, what type of the surgeries were included? Not specified by the authors except for generic terms such as maxillary advancement or mandibular advancement. (See now page 17)
  7. Table 1 B: what does 0.629 +/- 4.652 mean? NLA angle change in degree? Changed
  8. Table 1C: “MMP 106.24”, the symbol of degree is missing here. Changed
  9. Table 1E, line 3, what’s the results about the NLA change for this study? Changed
  10. The authors considered sex and age for each study, but the information of sex and age are listed separately with the NLA changes. And they are not discussed in the result section. Do the authors observed any differences on NLA change between genders or among different age groups? We added the reasons in the Results section on now page 6.
  11. What about the race distribution? Different population have different thickness of the soft tissue, so their upper lip response differently to same treatment strategy. We added the rationale for not including race under the Results section on page 6.
  12. Page 15 discussion section “Authors should discuss the results and how they can be interpreted from the perspective of previous studies and of the working hypotheses. The findings and their implications should be discussed in the broadest context possible. Future research directions may also be highlighted.” These sentences are from MDPI template instruction. Same on page 17, the first sentence for conclusion section. Yes, that was an interference during the online upload of the article. We corrected it.
  13. The discussion section is a whole long paragraph which repeats the research findings from other studied. It is reader unfriendly and it has no authors true discussion points. We made several changes to make it more reader friendly and we expanded the conclusion as well. We hope we improved its ease of reading.

Thank you so much for your very good points and observation.

Round 2

Reviewer 1 Report

The manuscript has been improved considering the reviewers suggestion. Tables should be adjusted regarding format and dimensions. Grammar and spacing should be checked once more. 

Author Response

To the Editor in Chief of                            

Applied Sciences 

Thursday 04 2021

Dear Editor,

I'm sending the revised manuscript entitled “Evaluation of the Nasolabial Angle in Orthodontic Diagnosis : A Systematic Review “

Thank you for considering our article for publication. We were pleased to receive the generous comments of the reviewers on our manuscript.  In the revised manuscript, we have carefully considered reviewers’ comments and suggestions. 

The reviewer’s comments were very helpful and we are appreciative of such constructive feedback on our submission.

The review comments are in black , while the author response is in red

The manuscript has been improved considering the reviewers suggestion. Tables should be adjusted regarding format and dimensions. Grammar and spacing should be checked once more. 

We had a reviewer whose mother tongue is American English read and edit the entire article, although unfortunately the little corrections were not highlighted in color. 

Unfortunately, it seems that things shift and disappear when we change platforms among our team members and the article’s uploading system. But we reviewed again formatting and spacing. Thank you.

Please note that we made some changes to the Keywords to better reflect the content of the article.

We believe that the manuscript is now suitable for publication on your Journal. 

Dr.  Sabina Saccomanno

On behalf of all the authors. 

Reviewer 2 Report

The authors have made extensive modifications to the revised submission. However, there are still some issues that need to be addressed before publication.

  1. figure 1: "records identified through ..." The last line of this box is cut off in the current submission. And the font throughout figure 1 is not consistent
  2. table 1: the degree symbol is missing in several locations. For example, the last line on page 6, under "EXTR." and "Distalization".
  3. The inclusive of study "CLINICAL AND RADIOGRAPHIC EVALUATION OF MAXILLARY CENTRAL INCISORS EXPOSURE IN PATIENT UNDERGOING MAXILLARY ADVANCEMENT" is not proper. In this paper, the authors only measured preop NLA, not postop NLA, as presented in their table 1. The statement "INCREASE IN NLA OF 1.20° AFTER MAXILLARY ADVANCEMENT" is from the literature discussion of this publication "The literature shows an increase in the nasolabial angle of 1.20° with anterior repositioning of the maxilla and a mean value of 0.65° for every 1 mm of advancement, although there is a wide variation in tissue response, some patients show an increase,9,15,19 while others show a reduction12 in the postsurgical period. Bundgaar, Melsen, and Terp7 hypothesized that an angular change may be related to muscle function on the site of the osteotomy.7 " Thus, I highly suggest the authors reevaluate the studies included in the current study and make proper statements and citations.

Author Response

To the Editor in Chief of                            

Applied Sciences 

Thursday 04 2021

Dear Editor,

I'm sending the revised manuscript entitled “Evaluation of the Nasolabial Angle in Orthodontic Diagnosis : A Systematic Review “

Thank you for considering our article for publication. We were pleased to receive the generous comments of the reviewers on our manuscript.  In the revised manuscript, we have carefully considered reviewers’ comments and suggestions. 

The reviewer’s comments were very helpful and we are appreciative of such constructive feedback on our submission.

The review comments are in black , while the author response is in red

The authors have made extensive modifications to the revised submission. However, there are still some issues that need to be addressed before publication.

  1. figure 1: "records identified through ..." The last line of this box is cut off in the current submission. And the font throughout figure 1 is not consistent Thank you for pointing that out. We tried to fix this problem which may be due to formatting shifts when changing platform. 
  2. table 1: the degree symbol is missing in several locations. For example, the last line on page 6, under "EXTR." and “Distalization”. We corrected that, thank you. 
  3. The inclusive of study "CLINICAL AND RADIOGRAPHIC EVALUATION OF MAXILLARY CENTRAL INCISORS EXPOSURE IN PATIENT UNDERGOING MAXILLARY ADVANCEMENT" is not proper. In this paper, the authors only measured preop NLA, not postop NLA, as presented in their table 1. The statement "INCREASE IN NLA OF 1.20° AFTER MAXILLARY ADVANCEMENT" is from the literature discussion of this publication "The literature shows an increase in the nasolabial angle of 1.20° with anterior repositioning of the maxilla and a mean value of 0.65° for every 1 mm of advancement, although there is a wide variation in tissue response, some patients show an increase,9,15,19 while others show a reduction12 in the postsurgical period. Bundgaar, Melsen, and Terp7 hypothesized that an angular change may be related to muscle function on the site of the osteotomy.7 " Thus, I highly suggest the authors reevaluate the studies included in the current study and make proper statements and citations.

We are grateful for your accurate revision and yes, you were right. This inclusion was an honest mistake due to confusion with data. We reviewed the original article and decided to exclude it from this systematic review. Therefore all the data in our article has been changed to reflect this exclusion.

We hope that after this last revision the article finally meets your standard for publication.

Please let us know.

We believe that the manuscript is now suitable for publication on your Journal. 

Dr.  Sabina Saccomanno

On behalf of all the authors. 

Round 3

Reviewer 2 Report

all comments are addressed.